# Enhanced sensitivity for electron affinity measurements of rare elements

F. M. Maier [1,2,3,4] ✉, E. Leistenschneider [2,5] ✉, M. Au [2], U. Bērziņš [6], Y. N. Vila Gracia[2], D. Hanstorp [7], C. Kanitz [2], V. Lagaki[1,2], S. Lechner [2], D. Leimbach[2,7], P. Plattner [2], M. Reponen [8], L. V. Rodriguez [2,9], S. Rothe [2], L. Schweikhard [1], M. Vilen [2], J. Warbinek [7,10,11] & S. Malbrunot-Ettenauer [2,3,12]

The electron affinity (EA), the energy released when a neutral atom binds an additional electron, is a fundamental property of atoms that is governed by electron-electron correlations and is strongly related to an element's chemical reactivity. However, conventional techniques for EA determination lack the experimental sensitivity to probe very scarce samples. As a result, the EA for the heaviest elements of the periodic table is entirely uncharted. Here, we present a novel technique to determine EAs through Laser Photodetachment Threshold Spectroscopy, performed in an electrostatic ion beam trap to increase the samples' exposure to laser photons and, thus, improve the experimental signal sensitivity by three orders of magnitude. Moreover, the additional exposure time allows the use of lower-power continuous-wave narrow-band lasers that reduce uncertainties associated with broadening effects induced by the laser bandwidth. By applying this technique, we measure the EA of $^{35}$Cl to be 3.612720(44) eV, achieving state-of-the-art precision while employing five orders of magnitude fewer anions. The demonstrated sensitivity paves the way for systematic EA measurements across isotopic chains - including isotope shifts and hyperfine splittings - and ultimately for the first direct determination of electron affinities in superheavy elements.

Unveiling the intricate electronic structure of atoms, including understanding how their attributes define the macroscopic chemical properties, was one of the greatest scientific achievements of the last century. Still even today, the pursuit of knowledge persists, aiming to push towards the limits of these systems' existence - for example, in superheavy elements - and to comprehend how their properties evolve as we edge closer to these boundaries[1]. The electron affinity (EA) is one of these fundamental attributes. It is defined as the energy released as an electron is attached to a neutral atom and, therefore, is strongly related to how the atom forms chemical bonds by sharing electrons[2,3]. The binding of such an additional electron is governed by complex electron-electron correlations[4]. Hence, EAs are powerful benchmarks for state-of-the-art atomic model frameworks, especially those reliant on many-body quantum methods[5,6]. Despite its fundamental importance, the EAs of several elements – especially those that are rare, heavy, and radioactive – are experimentally unknown[2].

[1]Institute of Physics, University of Greifswald, Greifswald, Germany. [2]CERN, Geneve, Switzerland. [3]TRIUMF, Vancouver, BC, Canada. [4]Facility for Rare Isotope Beams, East Lansing, MI, USA. [5]Nuclear Science Division, Lawrence Berkeley National Laboratory, Berkeley, CA, USA. [6]Institute of Atomic Physics and Spectroscopy, University of Latvia, Riga, Latvia. [7]Department of Physics, University of Gothenburg, Gothenburg, Sweden. [8]University of Jyväskylä, Jyvaskyla, Finland. [9]Max Planck Institute for Nuclear Physics, Heidelberg, Germany. [10]Johannes Gutenberg-Universität Mainz, Mainz, Germany. [11]GSI Helmholtzzentrum für Schwerionenforschung GmbH, Darmstadt, Germany. [12]University of Toronto, Toronto, ON, Canada. ✉e-mail: maierf@frib.msu.edu; erichleist@lbl.gov

State-of-the-art methods for high-precision EA measurements include Photodetachment Microscopy[7,8], Velocity Map Imaging (VMI)[9], and Laser Photodetachment Threshold (LPT) spectroscopy[10]. Their common measurement principle involves exposing negatively-charged ions to laser photons, resulting in the detachment of the extra electron from the subsequently neutral atom. In photodetachment microscopy and VMI, an anion beam is intersected by a perpendicularly propagating laser beam and, upon successful photodetachment, the outgoing electron is detected. To access the EA, the former method takes advantage of quantum interference effects, visualized as spatial electron patterns at the detector surface[11]. In VMI, the electron's velocity is mapped onto a position-sensitive detector, allowing the determination of its kinetic energy $E_e$ and, thus, of the EA according to $E_A = E - E_e$, where $E$ denotes the photon energy.

To overcome limitations of the crossed-beams configuration in terms of sensitivity, LPT spectroscopy employs a scheme in which laser and anion beams are collinearly overlapped. This enables a longer anion-laser interaction time, while simultaneously compressing the anions' velocity spread such that Doppler broadening is reduced, leading to improved spectroscopic resolution[12]. By observing the number of neutralized atoms as a function of the photon energy around the element's EA, the energy above which the anion is neutralized can be identified. However, the closer the threshold is approached, the more the photodetachment cross section diminishes[13]. As a result, even an LPT measurement requires access to substantial sample sizes, despite its advantage of extended laser-ion interaction time.

Enhancing the sensitivity of the LPT technique further would finally enable photodetachment studies for samples that are challenging to produce or isolate. For instance, measurements of photodetachment cross sections in molecular anions are required to understand their role and abundance in the interstellar medium[14–16]. Moreover, experimental determinations of the EA of molecules and atoms (including isotope-resolved measurements) could provide stringent benchmarks for theoretical methods in atomic many-body calculations, with implications across atomic and nuclear physics[17], as well as in quantum chemistry. The latter are, among others, valuable for predicting the structure and laser-coolability of (negatively-charged) molecules in contexts such as antimatter research[18–22] and radioactive molecules[23], which have recently emerged as promising probes for physics beyond the Standard Model of particle physics[24,25]. However, experimental data remain scarce, especially for samples that exist on Earth only in trace amounts or must be produced artificially - despite their relevance to several fields, including radio-pharmaceuticals research[26,27] and fundamental atomic theory. For example, since the atomic shell stabilization due to relativistic effects grows with increasing atomic number, EA measurements of actinide and superheavy elements would challenge the predictive power of fully-relativistic many-body quantum models[4–6]. They would also provide critical tests of the limits of periodicity in the table of elements[5,28,29], with oganesson ($Z = 118$) as the extreme case – predicted to be the first noble gas with a positive EA[30], and therefore capable of forming a stable anion.

As a proof of concept, the GANDALPH collaboration recently succeeded in accessing the EA of artificially produced $^{128}$I (ref. 31) and, in a subsequent milestone, measured for the first time the EA of a radioactive element, astatine[3]. This was achieved by employing high-efficiency, low-background neutral particle detectors[32] and a beam of astatine anions of $\approx 600$ fA ($3.75 \cdot 10^6$ particles per second), produced artificially by the radioactive ion beam (RIB) facility ISOLDE[33] at the European Organization for Nuclear Research (CERN). The successful EA determination, despite such a low anion intensity, underscores the potential of the LPT technique for analyzing rare samples. However, even with the collinear anion-laser overlap used in conventional LPT approaches, such as employed by GANDALPH, each anion is exposed

to the laser for less than a few microseconds before being discarded, resulting in a negligible fraction of anions undergoing photodetachment. Thus, such a single-pass approach cannot efficiently utilize the most scarcely produced samples, e.g., the actinides and superheavy elements, where production yields are at or even below just a few anions per second.

A pathway to higher efficiency is to increase the probability for anion-laser interaction by confining the anions in an ion trap. While this approach has been employed in radiofrequency[14,15,34,35] and Penning traps[36–38] for decades, it has limitations. The quasi-stationary motion of stored anions results in a lack of control over the generated neutral atom, leading to a low detection efficiency and hence a persistent reliance on large anion ensembles. In our ambition to overcome this limitation, we draw inspiration from a recent LPT measurement conducted in a storage ring. At the Double Electrostatic Ion Ring ExpEriment (DESIREE)[39,40], a beam of oxygen anions was repeatedly probed by a laser along a section of the revolving anions' trajectories[41]. The resulting neutral atoms maintained their momentum and left the storage ring along a well-defined path towards a detector, resulting in a high detection efficiency compared to previous studies in ion traps. This approach allowed the use of low-power narrow-bandwidth continuous wave (cw) lasers, enabling the highest precision in an EA measurement to date. However, the anion beam intensity used was several orders of magnitude higher than what is available for exotic species and, therefore, not compatible with experimental requirements of RIB facilities, the only places where short-lived radioactive species can be produced.

Here, we present a novel technique for the precise determination of EAs, tailored to effectively utilize rare samples produced in RIB facilities, while also leveraging the ion-beam storage concept pioneered at DESIREE for these studies. Our approach involves LPT experiments within an electrostatic ion beam trap, also referred to as a Multi-Reflection Time-of-Flight (MR-ToF) device[42,43]. Within these traps, anions are confined between a pair of electrostatic mirrors, separated by a field-free drift region where they travel at a constant velocity. For the present experiment, they are collinearly illuminated by a spectroscopy laser. This method is an extension of the Multi Ion Reflection Apparatus for Collinear Laser Spectroscopy (MIRACLS) technique[44,45], which enhances experimental sensitivity by exposing confined ions to lasers for longer durations. It allows for repeated laser probing of rare species while maintaining the high resolution offered by the collinear geometry[12]. Originally developed for fluorescence-based laser spectroscopy, the MIRACLS scheme is also particularly advantageous for LPT due to the efficient inertial guiding of neutralized atoms to an externally positioned detector, thus greatly increasing geometrical detection efficiency compared to other trap-based LPT methods. In the following, we demonstrate that our new approach achieves accurate EA determinations with competitive precision employing orders of magnitude fewer anions than conventional techniques, paving the way towards EA measurements at an one-atom-at-a-time sensitivity. The small footprint of MR-ToF devices, coupled with their reduced operational and maintenance requirements compared to storage rings, and their widespread use at RIB facilities as high-resolution mass spectrometers and mass separators[46–51] make our approach particularly compelling for LPT in rare-isotope research.

## Results
### Development of the experimental technique
To showcase the innovative nature of our new technique, we chose chlorine (Cl) for an EA measurement with improved experimental sensitivity. $Cl^-$ stands out as the most tightly bound anion of any element in the periodic table, much like helium's unique position among neutral atoms. This strong binding allows $Cl^-$ to be easily produced in specialized sources with the added simplicity of not exhibiting any excited state in its anion. Hence, Cl's EA is well investigated both

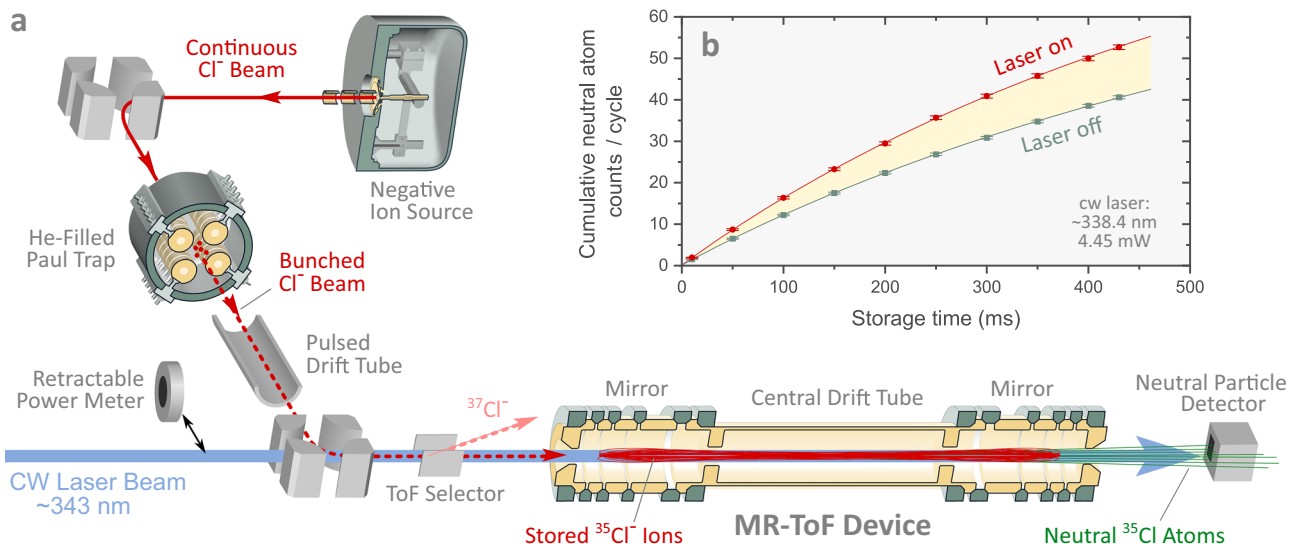

**Fig. 1 | Schematic of the experimental setup. a** A continuous beam of Cl⁻ anions produced by an ion source is accumulated in a He-filled Paul trap, from where anions are sent downstream in low-emittance bunches. After kinetic energy adjustment and isotope selection by time-of-flight, $^{35}$Cl⁻ anions are stored in an MR-ToF device, where they can be illuminated by a collinear laser beam. Atoms produced by neutralization processes leave the MR-ToF instrument and are registered by a detector. **b** The cumulative signal of neutral particles impinging on the detector grows with storage time. By measuring the difference (yellow band) between the cumulated neutral atom counts per cycle with (red circles) and without (green squares) laser illumination, the laser-induced photodetachment signal is determined. Error bars represent $1\sigma$ uncertainty, and lines are guiding the eye.

theoretically[13,17,52] and experimentally[37,53,54] and thus serves as a suitable case for benchmarking novel methods. Our primary objective is to quantify the number of anions required, comparing our method's sensitivity with that obtained with conventional approaches, while maintaining the highest level of experimental precision.

The measurement is performed with MIRACLS' low-energy MR-ToF setup[44,55–60] located at ISOLDE/CERN, which has been modified for the present photodetachment measurements. A schematic layout of the experimental setup is shown in Fig. 1a. A continuous beam of chlorine anions containing both stable isotopes of chlorine, $^{35}$Cl and $^{37}$Cl, is produced by a negative surface ion source[61]. The beam is delivered to a Paul trap, where the anions are captured, accumulated, and cooled via collisions with room-temperature helium buffer gas. Subsequently, the anions are released from the trap in low-emittance bunches and sent towards the MR-ToF device. In this process, the anion bunches pass through two high-voltage switching elements: a pulsed drift tube, which adjusts the anions' kinetic energy, and a deflector, which isolates the $^{35}$Cl⁻ isotope by time-of-flight selection, ensuring that the measurement pertains solely to the electron affinity of $^{35}$Cl⁻.

Once the $^{35}$Cl⁻ anions arrive at the center of the MR-ToF's central drift tube (CDT), its electric potential is switched to capture the anion bunch[62]. As a result, the anions - propagating with a kinetic energy of approximately 1.4 keV - are repeatedly reflected between two electrostatic mirrors and thus confined over a length of about 0.35 m. The stored anions are collinearly overlapped with a 1 to 5 mW cw laser beam, which is scanned in the region around 343 nm to probe the photodetachment threshold. A laser power meter can be inserted into the laser path just in front of the vacuum-air interfacing Brewster window. As stored anions are neutralized, either by laser photodetachment or collision-induced detachment with residual gas particles at a pressure of $3 \cdot 10^{-8}$ mbar in the present MR-ToF device, they escape as neutral atoms and are detected by a MagneTOF Mini Detector from ETP Ion Detect placed outside the ion trap. After typically half a second of anion storage, corresponding to approximately 60 thousand revolutions, the remaining $^{35}$Cl⁻ anions are extracted from the MR-ToF device. The detector is left to acquire background data on events unrelated to anion storage for an equivalent duration, after

which the measurement cycle is completed, and a new cycle restarted. More details on the experimental procedures can be found in the Methods section.

To avoid direct laser incidence onto the active area of the detector, which would generate a large background of photoelectrons, the neutral atom detector is slightly displaced off-axis. As a consequence, only about 40% of the atoms that arrive at the detector plane hit its active area. The integrated counts of neutral atoms per cycle ($N$) are determined by the number of counts observed while the MR-ToF device is loaded with anions, subtracted by the number of counts registered while acquiring background data after the anions are extracted. This component of the background, originating from detector dark counts and photoelectrons that are emitted when laser (stray) light hits the detector, accounts for about 3% of the total counts per cycle. The detector dark counts amount to $\approx 0.4$ counts per cycle and the laser-induced background counts are $\approx 0.2$ counts per cycle for each milliwatt of laser power.

As evident in Fig. 1b, the cumulative neutral atom counts per cycle grow with longer anion storage times, indicating a significant improvement in experimental sensitivity with longer laser exposure. Eventually, the growth rate diminishes as the population of anions is depleted due to neutralization. To determine the fraction of the integrated counts per cycle due to laser-induced photodetachment, measurements with laser illumination are interleaved by identical measurements with the laser off. As described in detail in the Methods section, the photodetachment signal is then defined as $S = N_{\text{on}} - N_{\text{off}}^{\text{int}}$, where $N_{\text{on}}$ is the number of counts of neutral atoms per cycle detected with lasers on, and $N_{\text{off}}^{\text{int}}$ is the interpolated (int) counts of atoms detected between the two interleaving measurements with laser off. To properly compare measurements taken at distinct laser powers and total stored anion content, we employ the photodetachment signal strength ($\mathcal{S}$), defined as the relative, power-normalized excess counts per cycle of neutral atoms produced with laser illumination:

$$\mathcal{S} = \frac{S}{N_{\text{off}}^{\text{int}} \cdot P}. \tag{1}$$

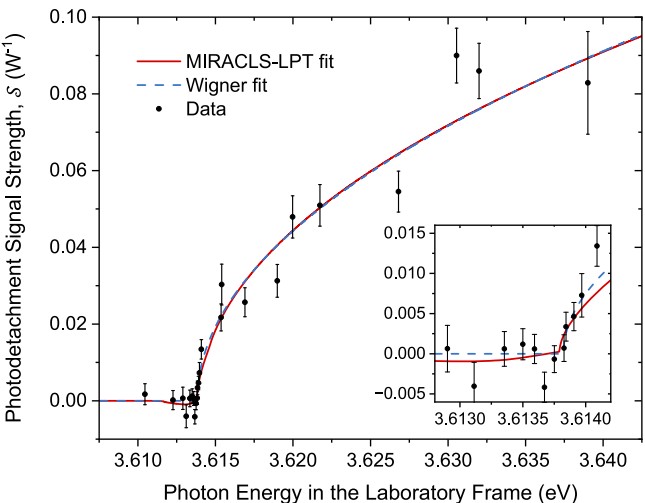

**Fig. 2 | The LPT spectrum of $^{35}$Cl.** Measured photodetachment signal strength $\mathcal{S}$, as defined in Eq. (1) (black dots), as a function of laser photon energy. Two models are fitted to the data: the Wigner threshold function (Eq. (2), blue dashed line) and a more realistic model adapted to the MIRACLS approach (red line). A zoom of the region around the threshold is shown in the inset. The photon energy is presented in the laboratory frame, i.e., not Doppler corrected. Error bars represent $1\sigma$ uncertainty.

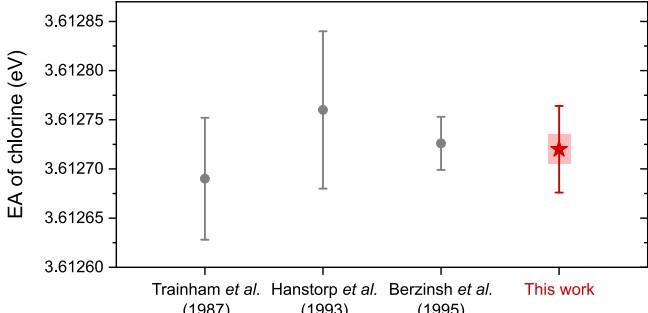

**Fig. 3 | Electron affinity $E_A$ of the chlorine atom as reported from several previous experiments**[37,53,54] **(gray circles) compared to the value obtained in the present work (red star).** Error bars represent $1\sigma$ total uncertainties, while the pink band denotes the $1\sigma$ statistical contribution to the uncertainty in our work. The systematic uncertainty is almost entirely due to a blueshift depletion when exploiting our MIRACLS scheme in collinear geometry, but would be absent in an anticollinear measurement configuration. See text for details.

Here, $P$ is the average laser power and $N_{\mathrm{off}}^{\mathrm{int}}$ is used as a proxy for the total anion content, since it is directly proportional to the number of stored anions (see Methods section).

The laser wavelength is scanned between about 340.5 and 343.5 nm, in search of the threshold above which a photodetachment signal is found. In contrast to most LPT experiments, we approach the threshold through successive approximations rather than with uniform steps, with higher statistics acquired closer to the threshold. This method focuses on the acquisition where data is more relevant, so a useful spectrum can be obtained with the least experimental time possible. The resulting LPT spectrum is shown in Fig. 2.

### The electron affinity of chlorine

The LPT spectrum relates to the photodetachment cross section ($\sigma$) which is described by the Wigner threshold law[63]:

$$\sigma(E) = \begin{cases} 0, & E \le E_A \\ A_w \left(E - E_A\right)^{l+1/2}, & E > E_A \end{cases} \quad (2)$$

where $E_A$ is the atom's EA, $E$ is the photon energy in the anions' rest frame, $l$ the orbital angular momentum quantum number of the detached electron, and $A_w$ is a scaling factor. The $^{35}$Cl$^-$ anion has a pure $3p^6\ ^1S_0$ configuration which does not exhibit any fine nor hyperfine splitting. The outgoing electron, as it is photodetached from a valence $p$ orbital, has either $l = 0$ or $l = 2$ according to the selection rules. The latter, however, is strongly suppressed close to the threshold due to the centrifugal barrier[4], permitting us to neglect it and to assume a pure $s$-wave emission. Finally, after the photodetachment process, the ground state of the remaining neutral atom has a configuration of $3p^5\ ^2P_{3/2}$, which splits into four hyperfine states as the $^{35}$Cl nucleus has a spin of 3/2. The maximum energy level difference between the hyperfine states is about 4.2 $\mu$eV[64], which is unresolvable given the steps in laser wavelength of the present experiment. Therefore, we approximate the photodetachment process in this experiment to occur between a pure two-level system.

The threshold position is determined by fitting Eq. (2) to the data, with the resulting curve shown as a dashed blue line in Fig. 2. The corresponding EA is extracted by applying the appropriate Doppler

shift (see Methods section). The MIRACLS approach derives its strength from repeated probing of anions, which introduces additional considerations in the application of the threshold law. The two most relevant are the continuous Doppler shift and the blueshift depletion. The first refers to the non-monochromatic velocity distribution of anions as they are decelerated and re-accelerated in the mirrors of the MR-ToF device. The second effect is caused by anions moving away from the detector but still overlapping with the laser and undergoing photodetachment. Since the detector is positioned to detect photodetachment events that happen in collinear, redshifted geometry (see Fig. 1), those events that happen in anticollinear, blueshifted configuration are not detected but still cause a depletion in the overall anion population. Thus, a negative normalized photodetachment signal $\mathcal{S}$ can arise at photon energies just below the threshold.

To account for both effects in the recorded MIRACLS data, an advanced fit function named MIRACLS-LPT is developed as shown in the Methods section. Its result, also presented in Fig. 2, is nearly identical to the Wigner law model except for details around the threshold. The extracted values of the EA are fully compatible with each other, see Methods section. Given the current level of experimental precision, the data itself does not discriminate between the two models. Therefore, we adopt the statistical uncertainty resulting from the Wigner model fit with 14 $\mu$eV and, conservatively, use the MIRACLS-LPT fit to estimate a potential systematic uncertainty arising from MIRACLS-specific factors, especially the blueshift depletion. This represents the dominant source of uncertainty in the final EA value, which - if detectable in the data at the given precision - arises from conducting the present experiment in a collinear geometry. This is due to the current experimental arrangement for neutral particle detection. If the measurement was conducted in anticollinear geometry, to be enabled by future advancements such as an efficient yet anion- and laser-transparent neutral particle detector or a laser-transparent detector combined with off-axis anion injection into the MR-ToF device, the final uncertainty could be reduced to levels comparable to those achieved with the Wigner function fit as the analogous redshift depletion effect would only occur at photon energies well above the threshold.

The present measurement results in an EA value of 3.612720(44) eV obtained from our MIRACLS-LPT fit. The 44 $\mu$eV total uncertainty combines statistical and systematic uncertainties, which are comprehensively characterized in the Methods section. Our value agrees with all previous measurements, see Fig. 3, with comparable uncertainty and hence contributes to an improved precision of the EA of $^{35}$Cl. Contrary to our experiment, however, the uncertainty of the previous results was dominated by the large spectral bandwidth of the pulsed lasers, which

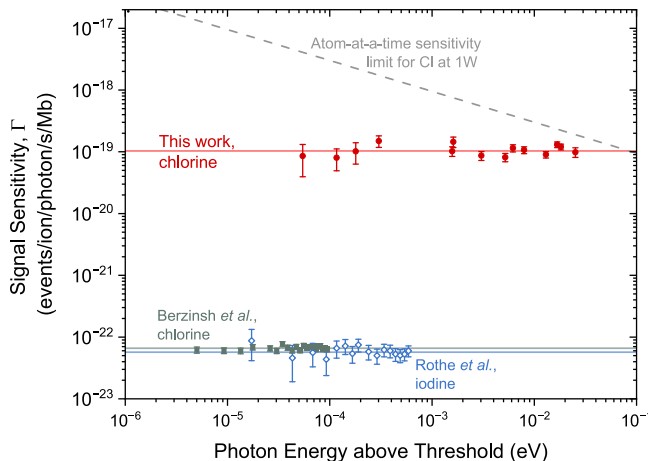

**Fig. 4 | Signal sensitivity of LPT experiments.** Signal sensitivity as a function of the photon energy above the threshold evaluated for this work (red circles), as well as for selected single-passage LPT measurements: Berzinsh et al.[53] on Cl (gray squares) and Rothe et al.[31] on I (blue open diamonds). The horizontal solid lines indicate the weighted averages for each experiment. The dashed line represents the signal sensitivity ideally required for one-atom-at-a-time LPT experiments, implying $S/N_{\mathrm{anion}} \approx 1$. Error bars represent $1\sigma$ uncertainty.

were required to produce the necessary intense photon flux. In our case, the prolonged laser exposure within the MR-ToF device compensates the reduced photon flux of the high-resolution cw laser employed, making laser bandwidth a negligible source of uncertainty. As such, our result demonstrates MIRACLS' capabilities of accurate EA measurements with competitive precision to conventional LPT methods. Most notably, beyond its contribution to the overall precision in chlorine's EA, this was accomplished with an anion sample five orders of magnitude smaller than in the previous work[53], as detailed in the following sections.

### Signal sensitivity

In experiments with rare radioisotopes, maximizing the use of each anion to generate experimental signals is essential. To characterize the performance of our technique under this criterion, we define the signal sensitivity

$$\Gamma = \frac{S}{N_{\mathrm{anion}} \cdot \Phi \cdot \sigma}, \tag{3}$$

in which $S = N_{\mathrm{on}} - N_{\mathrm{off}}^{\mathrm{int}}$ is the photodetachment signal introduced above, $N_{\mathrm{anion}}$ is the number of initially stored anions per cycle, $\Phi$ is the photon flux (expressed as number of photons per second), and $\sigma$ is the photodetachment cross section at the employed photon energy $E$ above the threshold.

In Fig. 4, we show $\Gamma$ for all measured points above the threshold in our experiment. $N_{\mathrm{anion}}$ is determined independently from the integrated counts/cycle of collision-induced neutralizations $N_{\mathrm{off}}^{\mathrm{int}}$ (see Methods section) and averages to 6100(800) anions/cycle initially stored in the MR-ToF device. $\Phi = P/E$ is obtained from the photon energy $E$ and laser power $P$ in each measurement, typically around 2 mW. The values for $\sigma$ are taken from ref. 13.

Our signal-sensitivity values are compared to those calculated from the data of two LPT experiments employing conventional single-pass approaches: on chlorine from ref. 53 and on iodine from ref. 31. Here, $N_{\mathrm{anion}}$, $N_{\mathrm{on}}$, and $N_{\mathrm{off}}^{\mathrm{int}}$ are given as the measured anion or atom number per second. As no dedicated background measurements were conducted, for $N_{\mathrm{off}}^{\mathrm{int}}$ the average number of detected neutral atoms for photon energies significantly below the photodetachment

threshold was taken. As evident in Fig. 4, the signal sensitivities $\Gamma$ for those measurements are comparable with each other. In contrast, our results represent an improvement of more than three orders of magnitude over single-passage experiments.

Our enhancement significantly advances LPT techniques, bringing them remarkably close to the capability of probing rare samples produced at the one-atom-at-a-time level. Ideally, such an LPT measurement should be able to achieve $S/N_{\mathrm{anion}} \approx 1$ at a reasonable target photon energy above the threshold and at a technologically attainable laser power. Figure 4 illustrates the required signal sensitivity for one-atom-at-a-time experiments (gray dashed line), assuming 1 W of laser power at the corresponding wavelength and the theoretical photodetachment cross section for chlorine. At least within the energy range of 0.01–0.1 eV above the threshold, absolute photodetachment cross-sections to the lowest bound state are also known, either theoretically for $H^-$, $Si^-$, $Cr^-$ [65] or experimentally for $F^-$, $Cl^-$, $Br^-$, $I^-$ [13], $O^-$ [66], and $Au^-$ [34]. Interestingly, when the same photon flux is assumed, the one-atom-at-a-time sensitivity limits for all these elements are within a factor of ~10 of the one shown for $Cl^-$. Therefore, while the sensitivity target indicated by the gray dashed line in Fig. 4 is that for chlorine, it also provides a useful order-of-magnitude guidance for LPT studies of rare samples involving other species.

In particular, Fig. 4 indicates that with a respective upgrade in photon flux, our current apparatus has already achieved the necessary one-atom-at-a-time signal sensitivity — namely, the ability to successfully neutralize each available anion — at photon energies as low as ~0.1 eV above the threshold. Hence, in the case of rare samples such as accelerator-produced, short-lived (chlorine) isotopes, EA measurements with a precision of 0.1 eV are within reach, even when only a single anion is injected into the MR-ToF device at a time. At this level of precision and anion intensity, however, a limiting factor of our current MIRACLS implementation of LPT studies is the total efficiency in the detection of neutralized atoms. The total efficiency is estimated to be ≈1.5% given by the product of intrinsic detector (≈10%), geometrical (≈40%), as well as extraction efficiency for atoms neutralized in the MR-ToF device, leaving towards the detector (≈36%). The respective efficiencies were either obtained via measurements and/or ion-optical simulations. Hence, ongoing work focuses on better-suited detection schemes along with other advances to ultimately gain even higher signal sensitivity and EA precision, especially beneficial for rare samples (see Anticipated Performance Improvements).

## Discussion

While the previous section focuses on the effectiveness of generating an LPT signal from a rare sample, the signal-to-noise $S/N$ ratio is a critical characteristic of an experimental apparatus to isolate a signal from the background. In conventional LPT studies with pulsed lasers, the leading source of background arises from laser-photon-induced detector events but can be largely eliminated when selecting events based on the neutralized atoms' delayed time of arrival after the photon pulse, as originally proposed in ref. 67. The remaining background is primarily due to collision-induced neutralization of anions. This is also the dominant source of background $N_{\mathrm{off}}^{\mathrm{int}}$, and thus the noise, in our current implementation of LPT work at MIRACLS with $Cl^-$ anions (see below). For comparison, at a photon energy of $5 \cdot 10^{-5}$ eV above the threshold and using the respective full data sets, the measurements on iodine reported in ref. 31 achieved an $S/N$ ratio approximately 2.5 times higher than that of our MIRACLS-based measurement. Similarly, for the chlorine data presented in ref. 53, the $S/N$ ratio is about 300 times higher than ours.

However, when comparing $S/N$ ratios across different experimental campaigns, it is important to note that, generally, the signal $S/N$ ratio increases with the number of available anions. The photodetachment studies at MIRACLS are conducted with 6100(800) anions/s, whereas the measurement reported in ref. 53 benefited from

an anion rate of $\approx 9.5 \cdot 10^8$ anions/s, i.e., $\approx 150{,}000$ times higher. In both cases, the actual data collection period was limited to approximately one week, hence, remaining within the same order of magnitude. Considering that the resulting (statistical) precision on the EA is within a factor of 2, comparable for both measurements, the ~5 orders of magnitude lower anion number used for our studies implies that the sensitivity for EA measurements is superior at MIRACLS.

The advantage of our technique is the consequence of two key points. (1) By using the MR-ToF device, we 'recycle' all available anions which could not be laser-neutralized during their first passage. This feature is particularly important for photon energies just above the EA, where the probability of successful photodetachment is exceedingly low, reflecting the inherently small cross-sections in this energy range. Thus, in conventional LPT studies, the vast majority of anions remain unaffected during their brief passage through the laser-ion interaction region, typically lasting only a few microseconds. In contrast, the MIRACLS approach allows for repeated probing of the anions, with the interaction time extended in the present work to about half a second. (2) We exploit a cw laser with a much narrower linewidth than a pulsed laser in the conventional LPT technique. Thus, the resolution of our measurements is about two orders of magnitude better. In fact, due to the use of a broadband pulsed high-power laser, a signal was observed in ref. 53 even for average photon energies slightly below the threshold. This smearing-out effect around the threshold limits the measurement precision when forced to work with broadband lasers. In other words, the high value in the $S/N$ ratio above threshold in ref. 53 has to be put in perspective with the attainable precision.

In threshold measurements, it is difficult to deconvolute these two factors for their implications on a technique's sensitivity around the threshold. However, we do not observe the threshold in our probed frequency range if we only analyze the first anion passage through the MR-ToF device. Thus, when considering all 60 thousand revolutions, the performance is significantly improved in terms of precision and sensitivity on the EA. This is due to the increase in observation time and the capacity to work with high-resolution, yet lower power cw lasers; both of which are facilitated by the MIRACLS scheme to exploit an MR-ToF device for laser spectroscopy of rare samples. In summary, we have accomplished a comparable precision to the previous EA measurement of chlorine[53] despite 5 orders of magnitude smaller sample sizes.

## Anticipated performance improvements

Building on our successful advancements, we identify the following opportunities to further enhance the LPT performance at MIRACLS. First, our signal counts above background are presently limited by the relatively low photon flux. Our laser power is only 2 mW, which reflects limitations in our laser setup (see Methods). Generating the laser light at the fundamental frequency with a dedicated laser close to the frequency doubler will increase the available laser power to about 1 W. We also note that chlorine has the most tightly bound anion. Therefore, for other elements, LPT studies require lower photon energies, typically allowing for a higher photon flux. For example, the predicted EA of actinium corresponds to a laser wavelength of 3960 nm (ref. 2), which enables a tenfold increase in photon flux at the same laser power compared to the chlorine studies conducted in this work.

Second, we currently work towards a neutral-particle detector that is more adequate for the use of cw lasers and MR-ToF devices. Such a detector will collect a larger fraction of atoms approaching a geometrical efficiency close to unity, compared to ≈40% in the present configuration. Additionally, its inherent detector efficiency is expected to be increased from the current ≈10% to close to unity when adopting design principles of ref. 32. A new detector system, hence, promises a total efficiency improved by a factor of ≈25 as compared to the currently employed MagneToF detector. Being (largely) transparent to laser photons, these improved neutral particle detectors could be installed on both sides of the MR-ToF device, facilitating simultaneous LPT studies in collinear and anticollinear configurations, provided the neutral particle detector is also anion-transparent or an off-axis ion injection into the MR-ToF device is realized in a future setup.

Third, while the LPT signal can be improved via higher detection efficiency and photon flux, a reduction in background will also improve the $S/N$ ratio. As mentioned above, the largest source of background in our data is due to collisional-induced neutralization of $Cl^-$ anions. This is a consequence of the vacuum conditions in the present MR-ToF device, which is affected by the base-pressure in this region, but also due to He flow from the Paul-trap cooler-buncher. The addition of supplemental pumping capacity next to the MR-ToF device, as well as a differential pumping section towards the gas-filled Paul trap, is expected to improve the vacuum quality by about two orders of magnitude and, thus, reduce the collisional-induced neutralization further. These improvements will be particularly beneficial for LPT studies of elements with lower electron affinity, where the anions may be more prone to neutralization upon collisions with residual gas particles.

Together, these planned improvements have the potential to further enhance the LPT sensitivity of the MIRACLS scheme by more than three orders of magnitude. Although the present work has already demonstrated multi-order-of-magnitude gains, it may thus represent merely a springboard for ushering in a new era of highly sensitive and precise LPT studies.

## Outlook

We have performed Laser Photodetachment Threshold (LPT) spectroscopy of the chlorine isotope $^{35}Cl$ by exploiting for the first time a Multi-Reflection Time-of-Flight (MR-ToF) device for this purpose. By repeatedly laser-probing a beam of $^{35}Cl^-$ anions reflected between the two electrostatic mirrors of the MR-ToF instrument, a signal sensitivity has been achieved that is three orders of magnitude larger than that of conventional LPT studies, where anions pass through the laser-ion interaction region only once, compared to about sixty thousand times in our work. By leveraging this scheme in our Multi Ion Reflection Apparatus for Collinear Laser Spectroscopy (MIRACLS) and taking advantage of the narrow linewidth of a continuous wave laser, a measurement of the electron affinity (EA) of $^{35}Cl$ has been conducted. Our result is in perfect agreement with the literature and, despite using approximately a factor of $10^5$ fewer anions, achieves a level of precision fully competitive with previous work, demonstrating the sensitivity advantage of the MIRACLS approach.

Beyond contributing to an improved precision in the EA of $^{35}Cl$, our MIRACLS concept thus opens a path towards accurate LPT studies of very rare samples, including accelerator-produced radionuclides such as (super-)heavy elements, which are out of reach of conventional methods. The apparatus employed for the present development and measurements is currently being coupled to the Gas-Filled Separator[68] at Lawrence Berkeley National Laboratory. Ongoing upgrades involve an improved vacuum quality to eliminate collision-induced neutralization as our presently dominating source of background, an increase in photon flux, as well as the design of a more efficient neutral particle detector. As discussed, the combination of these measures has the potential to further enhance the signal sensitivity and to increase the signal-to-noise ratio by several orders of magnitude beyond the achievements demonstrated in the present work.

In addition to the envisioned measurements with (super)heavy elements in which the EA presently remains unknown, we also note the growing interest in negatively-charged atoms and molecules for fundamental symmetry and antimatter studies, where accurate LPT investigations will be valuable in characterizing the (molecular) anion. An experimental method like MIRACLS becomes especially appealing for this purpose when studying rare or radioactive molecules. Once the improvements to our technique mentioned above have been

implemented, the MIRACLS approach also provides opportunities to significantly increase the precision of EA measurements for stable isotopes and molecules, as well as to improve the resolution of table-top LPT experiments to resolve detailed EA phenomena such as iso-tope shifts and hyperfine splittings. Combining MIRACLS' improved precision and sensitivity can facilitate isotope-shift measurements in the EAs across long isotopic chains, including short-lived radio-nuclides, serving as high-quality benchmarks for state-of-the-art atomic theory.

## Methods

### Creation and preparation of anions

Chlorine anions are produced using an ISOLDE-MK4 negative surface ion source[61], adapted to the requirements of the present studies. In short, a reservoir containing solid potassium chloride (KCl) is resis-tively heated, causing neutral chlorine atoms to effuse into the ion source. This source consists of a tantalum transfer tube and a lantha-num hexaboride ($LaB_6$) surface ionizer pellet, both of which are resistively heated to about 1500 °C. Thermionic electrons also emitted from the hot $LaB_6$ surface are deflected with a 0.04 T permanent magnetic field and absorbed in a dedicated electron collector. Once surface-ionized anions pass through the electron collector, they are accelerated across a 240 V extraction potential to form a continuous $Cl^-$ beam, which is injected into a linear Paul trap. After a precisely controlled anion-loading period, a bias voltage applied to a deflector electrode redirects the anion beam away from the trap's entrance. Anions confined within the Paul trap are thermalized through colli-sions with helium gas at room temperature. After this buffer-gas cooling, the anions are released from the trap as a well-defined bunch, characterized by a typical width of 100 ns (FWHM). Once the anions' kinetic energy is adjusted and the $^{35}Cl$ isotope is selected via time of flight, they are sent towards the MR-ToF device.

### Laser system, wavelength, and power measurements

The 343 nm light used in the photodetachment of $^{35}Cl^-$ anions is gen-erated by a laser system similar to the one described in ref. 44. A diode-pumped solid-state laser (Spectra Physics 20-W Millennia eV) produces the 532-nm laser light that pumps a dye cw ring laser (Sirah Matisse 2 DS laser) operated with a 1 g/L solution of 4-(Dicyanomethylene)-2-methyl-6-(4-dimethylaminostyryl)-4H-pyran (DCM) dye dissolved in phenoxyethanol + ethylene glycol (1:1) mix. The 686-nm output light is coupled into a high-power, large-mode-area optical fiber (LMA-15 single-mode 15 $\mu m$ core fiber) and transported with a power efficiency of about 25–30% over about 25 m from the laser room into the MR-ToF laboratory. There, the second-harmonic at 343 nm is generated with a frequency doubler (Sirah Lasertechnik, Wavetrain 2).

A pair of lenses matches the focus of the frequency-doubled laser beam to the center of the MR-ToF device. Finally, the roughly 1 mm diameter laser beam is guided through a periscope and two irises to the Brewster window, through which it enters the vacuum chamber. The laser power is measured before and after each photodetachment measurement using a retractable power meter positioned in the laser path, just in front of the Brewster window. Depending on the selected wavelength, it varies between 1 and 4.5 mW, being typically 2.2 mW around the $^{35}Cl^-$ photodetachment threshold.

The laser's fundamental frequency is continuously monitored - typically every 16 ms - using a wavelength meter (Toptica, High-Finesse WS8-2). This meter is regularly calibrated with the output of a temperature-stabilized Helium-Neon laser (SIOS SL-04/A) at approxi-mately 633 nm. The absolute frequency of the fundamental laser is hence known within  ± 5 MHz accuracy, thus, to 10 MHz in the frequency-doubled light. The output frequency is stabilized by locking the dye ring laser to an external reference cell, which is operated in a side-of-fringe locking scheme. Remaining frequency drifts are addressed by correcting the laser cavity once the frequency has drifted over 50 MHz from the set point. For each photon-energy data point in the LPT spectrum, the laser frequency is given by the mean frequency measured during the entire exposure period. Its uncertainty is con-servatively taken as half of the interval between the 2.3% and 97.7% percentiles of the acquired distribution, which contains about 95.4% of all data. The overall relative uncertainty of the laser frequency and thus photon energy is on the order of $5 \cdot 10^{-8}$ in the vicinity of the photo-detachment threshold. Since the final relative uncertainty of the EA obtained from this experiment is  $-1 \cdot 10^{-5}$, the uncertainty associated with the wavelength determination is negligible.

### Data structure and reduction

During each measurement cycle, an anion bunch consisting of, on average, 6100 (800) anions is confined in the MR-ToF device for 520 ms, while neutral atoms reaching the detector are counted. Afterwards, anions that have not undergone neutralization are released from the trap and the detector is left to acquire background data without stored anions for another 520 ms, before a new cycle is restarted. Each measurement consists of typically 250 consecutive cycles, taken under identical measurement conditions.

The data of each measurement cycle can be divided into four characteristic time windows. The first, termed the initial window ($\Delta t_{initial}$), spans the first 45.7 $\mu s$ of data acquisition. This period contains a mixture of signals from poorly captured or non-captured anions and from neutralized atoms while the MR-ToF's central drift tube (CDT) is still amid switching. Due to these mixed origins and transitional states of the device, data from this time window are excluded from the analysis. Following this, the atoms window ($\Delta t_{atoms}$) covers the remaining time when the trap is closed, i.e., for 520 ms minus the initial 45.7 $\mu s$. During this phase, detected events in the neutral particle detector originate from forward-directed atoms, neutralized either by photodetachment or collisions with residual gas, as well as back-ground sources unrelated to the anion storage, i.e., detector dark counts and laser-induced photoelectrons.

The third window ($\Delta t_{anions}$), occurs once the remaining anions are extracted from the MR-ToF device. Here, anions released toward the detector produce a narrow signal that suffers from a significant detector dead time. Thus, data within this narrow 65.7 $\mu s$ window are also excluded from subsequent analysis. Lastly, the background win-dow ($\Delta t_{bkgd}$) provides a baseline for subtraction of background counts unrelated to anions. This window, recorded for the same duration as $\Delta t_{atoms}$ while no anions are stored within the MR-ToF device, captures events solely from detector dark counts and laser-induced photoelectrons.

For each measurement, the integrated counts of neutral atoms per cycle is determined by $N = [\Sigma(\Delta t_{atoms}) - \Sigma(\Delta t_{bkgd})]/C$, where $\Sigma$ is the integrated number of counts in the stated time window and $C$ is the number of cycles. The LPT signal ($S = N_{on} - N_{off}^{int}$) is defined as the excess of neutral atom counts per cycle observed during data taking with laser illumination ($N_{on}$) compared to when no laser is present ($N_{off}^{int}$). The signal strength ($\mathcal{S}$, Eq. (1)) is the relative laser-induced increase in neutral atom counts per cycle, normalized by laser power ($P$). $N_{off}^{int}$ is obtained by the interpolated counts of atoms per cycle detected between the two interleaving measurements with the laser off, $N_{off}$.

The laser power $P$ used in the determination of $\mathcal{S}$ is calculated as the average of the power measurements before and after each $N_{on}$ measurement. Its uncertainty is determined from the standard devia-tion of the two measurements, incremented by a global 6.2% relative uncertainty. This global uncertainty is added to account for instru-mental uncertainties of 5.0% as recommended by the manufacturer of the instrument, as well as an additional 3.6% to account for reading uncertainties and power fluctuations occurring during the laser-power

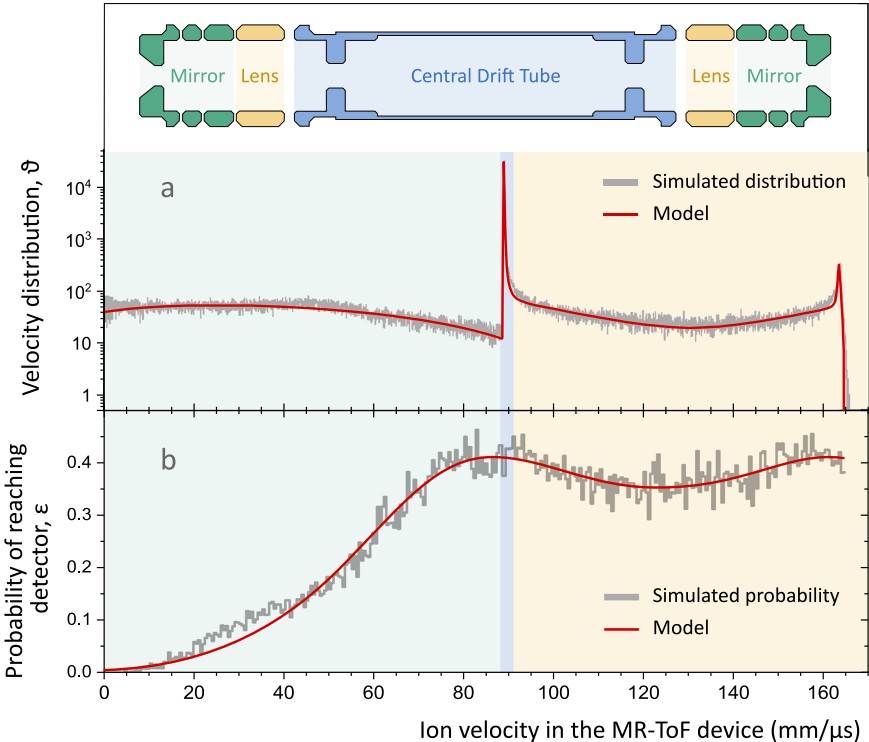

**Fig. 5 | Kinematics of anions confined in the MR-ToF device. a** Simulated distribution of anion velocities (gray histogram) and the model built upon it (red line). **b** Simulated probability distribution (gray histogram) representing the fraction of neutralized atoms with velocity $v$ moving toward the neutral particle detector, which reach the active surface of the detector, overlapped with the model built upon it (red line). In both panels, the color-shaded regions represent the different regions of influence of each electrode group in the MR-ToF instrument: Central Drift Tube (blue), Mirrors (green), and Lenses (yellow). A representation of the device's electrode structure is given for reference at the top of the figure.

measurement. The latter contribution is determined by the average standard deviation of several repeated readings made in the timescale of a typical laser-power measurement.

For each data point in the LPT spectrum, three to fifteen identical measurements with the laser on and with the laser off are interleaved. To obtain the final signal strength of each studied laser-frequency point, the weighted average of all individual determinations of $S$ is taken. Analogous procedures are followed for the determination of the signal sensitivity $\Gamma$.

## Modeling of velocity distribution, Doppler shift, as well as extraction and detection probabilities

In contrast to a well-defined, constant anion-beam energy in conventional, single-passage LPT studies, anions undergo repeated acceleration and deceleration throughout the course of the MIRACLS measurement process as part of their (longitudinal) confinement in the MR-ToF device. Therefore, an accurate understanding of the anions' velocity distribution is essential when evaluating the Doppler-shifted laser frequency in their rest frame while they overlap with the laser beam. This is achieved through simulations of the relevant anion trajectories combined with precise characterization of the electrical potential distribution throughout the setup.

The first requirement is to determine the kinetic energy of the stored anions when passing the central drift tube of the MR-ToF device. In a first approximation, this energy is given by the potentials $U$ applied to the trapping electrode of the Paul trap (PT), the energy-adjusting pulsed drift tube (PDT), and the spectrometer's central drift tube (CDT): $E_{kin} = |e(U_{PT} - U_{PDT} - U_{CDT})|$. However, any real high voltage (HV) switching circuit exhibits a non-vanishing time constant that can significantly impact the energy of the anions[45,60]. Moreover, the potential minimum at the Paul trap is only approximately given by $U_{PT}$. Hence, we

perform measurements of all electrostatic potentials applied to anion-energy-defining electrodes with high-precision voltage dividers and meters, as well as of the transient behavior of the Paul-trap extraction and the HV switching of the PDT and CDT. The results of these measurements are used for a time-evolving model of all electric potentials in the setup throughout a measurement cycle, which is fed into the ion-optical simulation code SIMION. Our simulation framework is described in ref. 45,55 and has been extensively benchmarked against experimental data[55,58-60]. In short, $^{35}Cl^-$ anions are thermalized in the Paul trap via collisions with 300 K buffer gas, extracted, guided onto the axis of the MR-ToF device, and stored for 100 revolutions. The resulting average kinetic energy of the anions in the CDT is found to be 1435.8 eV with a spread of 2.8 eV (FWHM) in the simulations. The value of the kinetic energy is used in our Wigner fit. A systematic uncertainty of 6.1 eV is assigned to the extraction of the mean anion energy, reflecting potential inaccuracies in the models of the HV switching elements. In the simulation, it is assumed that the anion bunch is precisely positioned in the center of the drift tubes when the HV switching is activated. Possible delays and offsets due to the experimental setup are also considered: a maximum 0.2 $\mu$s offset in switching time leads to a change in kinetic energy of approximately 0.1 eV, significantly smaller than the systematic uncertainty on the average kinetic energy.

In a second step, the ion-optical simulations are used to characterize the distribution of anion velocities $\vartheta(v)$ in the axial direction, parallel to the laser-beam axis. To this end, the kinetic energies $E_{kin}$ of 6000 simulated individual anions are sampled at random time intervals for 80 separate repetitions of the simulations while the $Cl^-$ anions remain confined in the MR-ToF device. Due to the transient behavior of the HV switch during anion capture, the sampling starts at revolution 50. The anion velocities $v$ are determined following the classical approximation $v = \sqrt{2E_{kin}/m}$. The uncertainty in EA determination due

to the use of this approximation instead of the relativistic formulation is as small as $4 \cdot 10^{-5}\,\mu eV$. Moreover, any transverse velocity component can be neglected at the current EA precision, compare to ref. 45.

In Fig. 5a, a histogram of the anion velocities is presented. The anions' nearly monochromatic kinetic energy while traveling through the CDT for a significant fraction of the revolution period results in a sharp velocity peak around 90 mm/μs. Additionally, the velocity distribution $\vartheta(v)$ features very broad, yet low-intensity tails on either side of the strong peak, corresponding to the time anions pass the MR-ToF's electrostatic mirrors and lenses, respectively, where the anions rapidly change their velocity. Therefore, the Doppler shift is expected to be predominantly governed by the sharp peak − justifying the use of the fixed-energy Wigner fit − while being complemented by long-range smearing from the tails of the velocity distribution. For a particular anion within this distribution, the photon energy $E'$ in the anion's reference frame is calculated according to

$$E'(v, E) = E \cdot \frac{1 \pm \frac{v}{c}}{\sqrt{1 - \left(\frac{v}{c}\right)^2}}, \tag{4}$$

where $c$ is the speed of light, $v$ is the velocity of the anion in the laboratory frame, and $E$ is the photon energy in the laboratory frame. The ± sign refers to anticollinear/collinear orientation.

Finally, neutralization events occur at various positions within the MR-ToF device and thus at different anion velocities, each associated with a distinct probability $\varepsilon(v)$ that the resulting neutral atom reaches the detector. To evaluate $\varepsilon(v)$, the same simulation framework is exploited, with anions being neutralized at random times during their flight through the MR-ToF device. The resulting neutral atoms retain the flight path and kinetic energy of the parent anions and propagate until colliding with a mechanical structure in the simulated setup. $\varepsilon(v)$ is evaluated as the fraction of all atoms with velocity $v$, moving toward the neutral particle detector, that strike its active surface. The results reveal that the probability of reaching the detector is about 40% if the neutralization happens anywhere in the MR-ToF's CDT and lens electrodes, but decreases monotonically as the anions move further into the mirrors, see Fig. 5b.

### The MIRACLS-LPT model

In a conventional LPT experiment, the form of the signal (on top of its background) has the shape of a Wigner threshold function, as outlined in Eq. (2). The MIRACLS approach introduces two major complications that may distort the shape provided by this model: the continuous Doppler shift (as illustrated in Fig. 5) and the blueshift depletion. A comprehensive interpretation of the experimentally detected neutral atoms for a fixed, monochromatic laser frequency in the laboratory frame must therefore consider: (1) the velocity distribution $\vartheta(v)$ and its associated Doppler shifts to the rest frame of the respective anions, (2) the probability $\varepsilon(v)$ of neutralized atoms reaching the neutral particle detector, and (3) the reduction in the available anion number per revolution due to anion neutralization in previous revolutions. To this end, we adopt the following model for the LPT spectrum, tailored to the MIRACLS approach in collinear geometry, which emulates the observed LPT signal strength as described in Eq. (1).

We model the neutralization of an initially stored anion population per cycle, $N_0$ called $N_{anion}$ in the main text, through the various available channels. The number of stored anions in a cycle $N_I$ evolves in time $t$ as $N_I(t) = N_0\, e^{-t \cdot \Sigma \lambda_i}$, where $\lambda_i$ represent decay rates of all available neutralization channels: collision-induced detachment (CID), collinear photodetachment (LPT), and anticollinear photodetachment ($\overline{LPT}$). Consequently, the total number of neutralized atoms per cycle $N$, integrated since $t = 0$, evolves in time as $N(t) = N_0 - N_I(t)$.

The counts per cycle observed at the neutral particle detector can then be modeled for the case with laser on ($N_{on}$), with all channels

open, and with laser off ($N_{off}$) with only the CID channel available:

$$N_{on}(t) = N_0 \left(1 - e^{-t \cdot (\lambda_{CID} + \lambda_{LPT} + \lambda_{\overline{LPT}})}\right) \left(1 - \frac{\lambda_{CID}/2 + \lambda_{\overline{LPT}}}{\lambda_{CID} + \lambda_{LPT} + \lambda_{\overline{LPT}}}\right), \tag{5}$$

$$N_{off}(t) = N_0 \left(1 - e^{-t \cdot \lambda_{CID}}\right) \frac{1}{2}. \tag{6}$$

In each equation, the first term in parentheses represents the fraction of anions still confined within the MR-ToF device at time $t$. This is followed by a term denoting the fraction of detectable atoms, i.e., excluding those which are lost through the side of the trap not covered by the detector. The latter correction appropriately accounts for the blueshift depletion. These equations can be substituted into Eq. (1) to obtain an expression for the signal strength. Taken at the end of the trapping cycle $t = t_{trap}$, such a signal strength becomes

$$S = \frac{1}{P} \cdot \left[ 2 \cdot \frac{\left(1 - \frac{\lambda_{CID}/2 + \lambda_{\overline{LPT}}}{\lambda_{CID} + \lambda_{LPT} + \lambda_{\overline{LPT}}}\right) \left(1 - e^{-[\lambda_{CID} + \lambda_{LPT} + \lambda_{\overline{LPT}}] \cdot t_{trap}}\right)}{1 - e^{-\lambda_{CID} \cdot t_{trap}}} - 1 \right]. \tag{7}$$

The CID rate $\lambda_{CID}$ is a function of the residual gas pressure in the MR-ToF instrument and can be considered static throughout the measurements. The photodetachment rates, $\lambda_{LPT}$ and $\lambda_{\overline{LPT}}$, are proportional to the photodetachment cross section ($\sigma$, Eq. (2)) and will provide the characteristic shape of the signal curve: $\lambda_{LPT}(E') \propto \sigma(E')$, where $E'$ is the photon energy in the anion's reference frame. According to Eq. (4), $E'$ depends on the anion velocity $v$ and the photon energy $E$ in the laboratory frame; consequently, $S = S(v, E)$ in Eq. (7). The LPT rates for the collinear or anticollinear cases have identical formulation but with opposite Doppler-shifted photon energies. Therefore, $\lambda_{LPT}$ employs Eq. (4) with a negative sign, while $\lambda_{\overline{LPT}}$ uses it with a positive sign.

To account for the continuous distribution of anion velocities in the MR-ToF device, we convolute Eq. (7) with the velocity distribution $\vartheta(v)$ of anions in the MR-ToF instrument, modeled after the simulated distribution presented in Fig. 5a, and the probability $\varepsilon(v)$ of neutralized atoms reaching the detector, also modeled after the same simulated results, see Fig. 5b:

$$S'(E) = \frac{\int \varepsilon(v) \cdot \vartheta(v) \cdot S(v, E)\, dv}{\int \varepsilon(v) \cdot \vartheta(v)\, dv}. \tag{8}$$

This approximation is appropriate when the number of anions neutralized by the laser per measurement cycle is negligible compared to the total number of stored anions, as is the case in our measurements near the photodetachment threshold. Despite its complexity, this final expression is a function of photon energy $E$ in the laboratory frame with only three free parameters: the rate $\lambda_{CID}$, as well as the threshold value $E_A$ and the scaling factor $A_w$ from Eq. (2).

### Uncertainties in EA determination

Table 1 compiles all sources of uncertainty in the determination of the $E_A$ of $^{35}Cl$ as described throughout this manuscript. A total uncertainty of 44 μeV is obtained by combining statistical and systematic contributions in quadrature. It is dominated by the systematic uncertainty associated with the incorporation of the blueshift depletion into the MIRACLS-LPT function, which is evaluated to be 42 μeV.

### Determination of the number of stored anions

While knowledge of the number of anions $N_0$ is not required for the determination of the electron affinity, it is essential to assess the signal sensitivity of the new MIRACLS methodology applied in this study. However, the number of anions is neither sufficiently high to determine it using electric current integration methods, nor low enough for

**Table 1 | Systematic and statistical contributions to the uncertainty in the determination of the EA of $^{35}Cl$**

| Contribution | Statistical uncertainty | Systematic uncertainty | |
|---|---|---|---|
| Fit uncertainty | 14 $\mu eV$ | 42 $\mu eV$ | in $E_A$ |
| Laser frequency | (≈100 MHz | 10 MHz | on frequency) |
| | ≈0.41 $\mu eV$ | 0.04 $\mu eV$ | in $E_A$ |
| Ion beam energy | | (6.1 eV | on beam energy) |
| | | 2.3 $\mu eV$ | in $E_A$ |
| Offset switch timings | | (0.1 eV | on beam energy) |
| | | 0.04 $\mu eV$ | in $E_A$ |
| Hyperfine structure | | 4.2 $\mu eV$ | in $E_A$ |
| Non-relativistic treatment of anion velocities | | $4 \cdot 10^{-5}$ $\mu eV$ | in $E_A$ |

single-ion counting techniques; i.e., the impact of the temporarily narrow anion bunch onto the detector results in a signal subject to a significant dead-time effect and, thus, delivers an unreliable result.

Instead, we utilize the number of neutral atoms produced by CID during the $N_{off}$ measurement cycles as a measurable proxy in the estimation of the initially injected total number of anions employed in the cycle, according to the model presented in Eq. (6). The number of neutral atoms is significantly more reliable since it is spread out over several hundreds of milliseconds and, thus, we always operate in a regime suitable for single-atom counting. The relationship between the counts of atoms per cycle $N_{off}$, and the number of stored anions in a cycle $N_I$ is investigated for different residual gas pressures. To this end, we perform several measurements of $N_{off}$ at decreasing quantities of anion injection into the MR-ToF device, until the anion signals arriving at the $\Delta t_{anions}$ time window are no longer suffering from significant dead time effects and single-ion counting of anions impinging on the detector is reliable. This procedure enables appropriate accounting for the detector's quantum efficiency for neutral atom particle detection in comparison to the efficiency for anions, well characterized by the manufacturer. Finally, anion and atom losses during transport to the detector are estimated using the aforementioned SIMION simulations and incorporated into the determination of detector geometrical acceptance, which is also employed in estimating $N_0$. Over the course of the $E_A$ measurements, $N_0$ varies between 3750 and 7460 initially stored anions per cycle, with an average of $N_0 = 6100(800)$ anions.

## Data availability

The data that support the plots within this paper and other findings of this study are available from the corresponding authors upon request.

## Code availability

Our unpublished computer codes used to generate results that are reported in the paper and central to its main claims will be made available upon request.

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

## Acknowledgements

The research leading to these results has received funding from the European Research Council (ERC) under the European Union's Horizon 2020 research and innovation program under grant agreement No. 679038 (S.M.-E.). The work of F.M.M. has been partly sponsored by the Wolfgang Gentner Program of the German Federal Ministry of Education and Research (grant no. 05E18CHA). E.L. acknowledges the support from the U.S. Department of Energy, Office of Science, Office of Nuclear Physics under contract number DE-AC02-05CH11231. M.A. and J.W. acknowledge support from the European Union's Horizon 2020 Research and Innovation Program (grant No. 861198, project LISA MSC ITN). U.B. was supported by the Fundamental and Applied Research Project (No. lzp-2023/1-0199) from the Latvian Science Council. This work is supported by the Swedish Research Council under the grant agreement No. 2020-03505 (D.H.), the CERN Budget for Knowledge Transfer to Medical Applications (S.M.-E.), and by the Natural Sciences and Engineering Council of Canada (NSERC) (S.M.-E.). TRIUMF receives federal funding via a contribution agreement with the National Research Council of Canada. We are grateful for the support of the MIRACLS project from CERN, the ISOLDE Collaboration, and the Max-Planck-Institut für Kernphysik (MPI K) in Heidelberg. We thank in particular K.

Crysalidis and Á. Koszorús for providing support in the laser setup and W. Nörtershäuser for all input and discussions.

## Author contributions

S.R., D.H. and S.M.-E. conceived the measurement idea for the present work from which E.L. and F.M.M. developed the detailed experimental program. F.M.M., E.L., M.A., U.B., Y.N.V.G., C.K., V.L., S.L., D.L., P.P., M.R., L.V.R., S.R., M.K.V. and J.W. prepared the experimental apparatus. The measurements themselves were conducted by F.M.M. and E.L. with laser support from L.V.R. The data analysis was performed by F.M.M. and E.L., in regular exchange with S.M.-E., and reviewed by S.R., D.H., and L.S. The manuscript was prepared by E.L., F.M.M. and S.M.-E. Resources and funding were secured by S.M.-E., L.S., D.H. and S.R. All authors discussed the results and provided comments on the manuscript.

## Funding

## Competing interests

The authors declare no competing interests.
