## [Transparent Peer Review file · Nature Communications]

Enhanced Sensitivity for Electron Affinity Measurements of Rare Elements

Corresponding Author: Dr Franziska Maier

Version 0:

Reviewer comments:

Reviewer #1

(Remarks to the Author)

This article reports a new measurement of the electron affinity of the (stable) isotope ³⁵-chlorine, using a new technique which achieves a much greater sensitivity than existing methods while attaining similar precision.

Electron affinities are fundamental atomic properties and continue to provide important benchmarks for atomic theory, of particular importance for the heaviest i.e., trans-lead species. As such species can only be synthesized using particle accelerators, they are produced in very limited quantities and in many cases, undergo transmutation with half-lives that are not necessarily compatible with certain measurement techniques.

The method presented in this work, collinear laser photo-detachment in a multi-reflection mass spectrograph, is new and innovative. Most importantly, the authors demonstrate a sensitivity improvement of three orders of magnitude over existing techniques, using a sample size that is five orders of magnitude smaller.

In my opinion, this manuscript demonstrates a major step forward for atomic physics and merits publication in Nature Communications. I believe there is one point – concerning the calibration of the measurement from charge-exchange cross-sections – that would merit more discussion (see below).

The authors give a short but perfectly sufficient review of the existing methods (Photo-detachment Microscopy, Velocity-Map Imaging and Laser Photo-detachment Threshold spectroscopy), which are relevant for the article. This is followed by a good description of their technique, with ample reference to published work detailing its characterization, as well as generous explanations in the “methods” section. The results are convincing: state-of-the-art accuracy – even improving the overall uncertainty of the ³⁵-chlorine electron affinity – with one hundred thousand times less ions. The key to their achievement is the many thousands of revolutions inside the ion trap, which greatly enhances the interaction of the ions with the laser beam. (Their method also allows using less laser power, which avoids the linewidth broadening present in single-pass experiments.) As such, it is the key to moving towards rare species and eventually on-line application for those having shorter half-lives.

Determining the number of trapped anions is essential and was done using a detailed ion-optics simulation and by measuring the rate of collision-induced dissociation due to the residual gas inside the trap with interleaved laser-on-off scans. The authors discuss this at length in the “Methods” section for chlorine, which has a well-known collision-induced dissociation cross-section. When extending this technique to “rare” species produced in limited quantities, aren’t the relevant cross-sections completely unknown? The whole point of the electron affinity work is to stimulate and test atomic physics theory but I wonder if the experimental result does not also depend on the collisional dissociation from the same theory. If this is relevant for extending the technique to rare species (mentioned in the title) then I believe some additional discussion would be warranted.

As the authors point out, chlorine is very tightly bound, meaning every element requiring investigation will be less so and likely suffer increased dissociation. Could the repeated accelerations in the device be an additional source of dissociation for certain (less-bound) species? I guess the future experiments will also require better vacuum (i.e. in the 10⁻¹⁰ range)?

Some editorial considerations for the authors:

Abstract: the explicit mention of the five-orders-of-magnitude reduction in sample size is mentioned in the same sentence as the improvement of sensitivity, qualified as "greatly improved". Why not also quote the (greater than) three-order-of-magnitude gain in sensitivity, nicely justified in Fig. 4? I might also suggest that the authors distinguish the additional gain brought by the ion "recycling" of the lower-power lasers, rather than grouping in the same sentence (which slightly diminishes the impact). For example: "Moreover, the additional exposure time allows the use of lower-power CW lasers that reduce uncertainties due to broadening."

Introduction – short suggestion for rewording the first sentence: "Unveiling the intricate electronic structure of atoms, including understanding how their attributes define the macroscopic chemical properties, was one of the greatest scientific achievements of the last century."

I also suggest modifying: "experimentally entirely unknown" to read: "experimentally unknown".

Reviewer #2

(Remarks to the Author)

The authors present the application of a new technique to improve the sensitivity of laser photodetachment threshold (LPT) spectroscopy. Currently LPT is one of the most precise method in determining the electron affinity of negative ions. However, due to the low cross section of the non-resonant photodetachment process the mainly used single pass LPT is hindered when determining the EA of rare or exotic species. The logical consequence is to go for a multi-pass measurement where the precious ions are probed multiple times and therefore increasing the sensitivity. Inspired from storage ring experiments the authors use a Multi-Reflection Rime Of Flight apparatus for this purpose. Keeping in mind that the previous storage ring measurements were performed with much higher ion beam intensities I disagree with the statement in the introduction arguing that storage ring measurements are not therefore not compatible with RIB facilities. Facilites like the ESR at GSI show that even single ion sensitivity can be achieved using storage rings, but undeniably under large infrastructure effort.

To present the validity of their approach the authors present a measurement of the EA of Cl-35 using the MR-TOF technique. The results gained are thoroughly analysed and the conclusion drawn are sound.

The proposed method can have a significant impact on future electron affinity determination of rare and exotic species. Being a key experimental accessible parameter of negative ions the electron affinity gives important insight in further properties of rare species like e.g. their chemical behaviour.

The data and methodology presented in the supplementary material describes reasonably well the experimental methods and the modelling of the parameters specially focused on the used MR-TOF device. A comprehensive discussion of the achieved sensitivity is given.

The list of references is exhaustive and all the relevant previous works relevant for the current article have been cited.

A suggested improvement would in my opinion be a more comprehensive discussion of the pros and cons of the presented technique in comparison to using storage rings. Both methods gain from the use of multi-pass instead of the low sensitivity of a single pass experiment. However, both complicating effects of 'continuous Doppler shift' and 'blueshift depletion' are intrinsic to a MR-TOF device and non-existent in a storage ring. The manuscript would gain from such a discussion and emphasise why the authors claim that their method is superior.

Reviewer #3

(Remarks to the Author)

This manuscript, "Enhanced Sensitivity for Electron Affinity Measurements of Rare Elements," introduces a significant advancement in measuring Electron Affinity (EA), addressing the critical challenge of measuring EA of rare and heavy elements, a fundamental atomic property that remains uncharted due to insufficient experimental sensitivity of conventional techniques. The authors combined Laser Photodetachment Threshold (LPT) spectroscopy with a Multi-Reflection Time-of-Flight (MR-ToF) device to create a highly sensitive new method. The technique involves generating and injecting chlorine anions ($^{35}\text{Cl}^-$) into the MR-ToF device, where they are trapped for about 0.5 seconds (60,000 revolutions). A continuous-wave laser repeatedly probes the anions during this time. The paper demonstrates a three-order-of-magnitude improvement in signal sensitivity over conventional methods, using approximately five orders of magnitude fewer anions and less laser power. This breakthrough is poised to have a significant impact on atomic physics, nuclear physics, and the chemistry of super-heavy elements. For unstable isotopes, such as the heaviest elements, conventional measurement methods are not applicable. That's why the technique reported in this manuscript will pave the way for atomic sciences in the study of superheavy elements.

I recommend minor revisions for publication, which will help the reader's understanding. The minor revision list is as follows.

Minor Comments

- Line 120: The statement "A continuous beam of chlorine anions containing both stable isotopes of chlorine, ^{35}Cl and ^{37}Cl , is produced..." is followed by "a deflector which isolates the ^{35}Cl isotope by time-of-flight selection". Please clarify if ^{37}Cl is ever actually used, or if its initial mention is purely contextual.

- Line 132: Regarding "collision-induced detachment with residual gas particles at a pressure of 3×10^{-8} mbar," it would be beneficial to provide context on whether this pressure is typical for ion beam traps or a specific challenge for the MIRACLS apparatus, e.g., compared to storage rings.

- Line 143: "Photoelectrons that are emitted when laser (stray) light hits the detector" are mentioned as a background source. Given that the detector is "slightly displaced off-axis" (Line 138) "to avoid direct laser incidence," please clarify the nature and significance of this "stray light" background relative to the detector's dark counts.
- Line 166: The statement " $^{35}\text{Cl}^-$ anion has a pure $3p^6 1S_0$ configuration which does not exhibit any fine nor hyperfine splitting" refers to the anion. However, Line 170 states, "the ground state of the remaining neutral atom has a configuration of $3p^5 2P_{3/2}$, which splits into four hyperfine m states...". To avoid confusion, explicitly specify that the lack of fine/hyperfine structure refers to the anion ground state before photodetachment.
- Line 217: The description " $\Phi = P/E$ is obtained from the photon energy E and laser power P in each measurement, typically around 2 mW". For clarity, it would be more precise to write $\Phi = P / (h\nu)$ or P/E_{photon} to indicate that E refers to photon energy explicitly.

Version 1:

Reviewer comments:

Reviewer #1

(Remarks to the Author)

Thanks for the informative replies and relevant modifications to the manuscript. I am happy to accept your work as is for publication - congratulations!

Reviewer #2

(Remarks to the Author)

The manuscript has undergone a minor revision and my review addresses only the changes compared to the original submission.

I appreciate that the authors have discussed the comparison of their approach to the applications of storage rings at RIBs and the single ion sensitivity of storage rings. I understand that a thorough discussion in the manuscript would require more space which is in contradiction to the word limit imposed by the journal. This discussion is therefore left for future research and upcoming publications.

The comments of the other reviews have also been addressed carefully resulting in a more consistent manuscript. I fully support the publication of the work in its present form.

Reviewer #3

(Remarks to the Author)

made.

Dear reviewers,

We would like to thank you for your careful reading and response to the manuscript. We are pleased that all of you consider our work of high quality and our manuscript well suited for publication in Nature Communications pending the revision of your comments, which we address below.

In the following, we provide detailed answers to all comments and questions received. We also indicate which respective changes we made in the manuscript. Your comments and questions are in “red” while our responses are in “black”.

Thank you very much,

Franziska Maria Maier on behalf of all co-authors

Reviewer #1 (Remarks to the Author):

This article reports a new measurement of the electron affinity of the (stable) isotope 35-chlorine, using a new technique which achieves a much greater sensitivity than existing methods while attaining similar precision.

Electron affinities are fundamental atomic properties and continue to provide important benchmarks for atomic theory, of particular importance for the heaviest i.e., trans-lead species. As such species can only be synthesized using particle accelerators, they are produced in very limited quantities and in many cases, undergo transmutation with half-lives that are not necessarily compatible with certain measurement techniques.

The method presented in this work, collinear laser photo-detachment in a multi-reflection mass spectrograph, is new and innovative. Most importantly, the authors demonstrate a sensitivity improvement of three orders of magnitude over existing techniques, using a sample size that is five orders of magnitude smaller.

In my opinion, this manuscript demonstrates a major step forward for atomic physics and merits publication in Nature Communications.

Thank you very much for your positive review!

I believe there is one point – concerning the calibration of the measurement from charge-exchange cross-sections – that would merit more discussion (see below).

We have not used any collision-induced detachment cross sections from literature for any calibration. For the determination of the electron affinity via the MIRACLS-LPT fit model, λ_{CID} is a free parameter fitted to the experimental data in Fig. 2, as written in line 703. Moreover, the determination of the number of ions is based on the measurement of neutral atoms as a function of storage time in the MR-ToF device for different residual gas pressures in the MR-ToF device and different numbers of injected ions. We then related the number of detected neutral atoms with the number of measured ions and the respective residual gas pressure present in the MR-ToF device. We agree that the sentence “*The behavior of λ_{CID} is calibrated independently of the present measurements at several residual gas pressures in the MR-ToF device, and the EA measurements are performed at a fixed pressure, thus at a well-characterized*”

λ CID.” could have been misleading, we hence removed it. The corresponding paragraph now reads: **“The number of neutral atoms is significantly more reliable since it is spread out over several hundreds of milliseconds and, thus, we always operate in a regime suitable for single-atom counting. The relationship between the counts of atoms per cycle N_{off} and the number of stored anions in a cycle N_{I} is investigated for different residual gas pressures. To this end, we perform several measurements of N_{off} at decreasing quantities of anion injection into the MR-ToF device, until the anion signals arriving at the Δt_{anions} time window are no longer suffering from significant dead time effects and single-ion counting of anions impinging on the detector is reliable.”**

The authors give a short but perfectly sufficient review of the existing methods (Photo-detachment Microscopy, Velocity-Map Imaging and Laser Photo-detachment Threshold spectroscopy), which are relevant for the article. This is followed by a good description of their technique, with ample reference to published work detailing its characterization, as well as generous explanations in the “methods” section. The results are convincing: state-of-the-art accuracy – even improving the overall uncertainty of the 35-chlorine electron affinity – with one hundred thousand times less ions. The key to their achievement is the many thousands of revolutions inside the ion trap, which greatly enhances the interaction of the ions with the laser beam. (Their method also allows using less laser power, which avoids the linewidth broadening present in single-pass experiments.) As such, it is the key to moving towards rare species and eventually on-line application for those having shorter half-lives.

Thank you very much for the positive review!

Determining the number of trapped anions is essential and was done using a detailed ion-optics simulation and by measuring the rate of collision-induced dissociation due to the residual gas inside the trap with interleaved laser-on-off scans. The authors discuss this at length in the “Methods” section for chlorine, which has a well-known collision-induced dissociation cross-section. When extending this technique to “rare” species produced in limited quantities, aren’t the relevant cross-sections completely unknown? The whole point of the electron affinity work is to stimulate and test atomic physics theory but I wonder if the experimental result does not also depend on the collisional dissociation from the same theory. If this is relevant for extending the technique to rare species (mentioned in the title) then I believe some additional discussion would be warranted.

Indeed, the collision-induced detachment cross sections of rare species might be fully unknown. In the present work, the purpose of determining the number of trapped anions is to compare the sensitivity of our new technique against the one of single-passage measurements. For the measurement of the electron affinity itself, the determination of the number of trapped anions is not needed. To stress this aspect more, we added the text in bold to the methods section: **“While knowledge on the number of anions N_{O} is not required for the determination of the electron affinity, it is essential to assess the signal sensitivity of the new MIRACLS methodology applied in this study.”** Please also see our response to your related question further above.

As the authors point out, chlorine is very tightly bound, meaning every element requiring investigation will be less so and likely suffer increased dissociation. Could the repeated accelerations in the device be an additional source of dissociation for certain (less-bound) species? I guess the future experiments will also require better vacuum (i.e. in the 10^{-10} range)?

Indeed anions of elements with lower electron affinity might be more prone to neutralization upon collisions with residual gas particles. It is fully correct that such measurements will benefit from improved vacuum quality in the MR-ToF device. The residual gas pressure in MIRACLS low-energy setup used in the present studies was about 3×10^{-8} mbar. As indicated in the manuscript, we recently moved the apparatus from CERN to Lawrence Berkeley Laboratory. It is now coupled to the Gas-Filled Separator where (super)heavy elements can be produced. As part of the necessary reconfiguration of the setup, first simple measures such as increasing the pumping capacity have already improved the vacuum quality by a factor of about 4.

Moreover, the MR-ToF device used here is a first-generation instrument. By further improvements in respect to the present state-of-the-art (such as a careful selection of components with further reduced outgassing rates, fully bakeable MR-ToF instrumentation, elimination of oil-based prevacuum pumps, the use of ion- and cryopumps, and multi-stage differential pumping to the He-filled Paul trap) a vacuum quality improved by up to two orders of magnitude compared to our present apparatus is expected to be within reach. This would be in line with what has already been achieved in other ion-trap or low-beam-energy installations of comparable complexity. Looking ahead, we therefore envision an entirely new setup dedicated to photodetachment measurements, with a strong emphasis on vacuum performance. In the case of a fully cryogenic MR-ToF device, the residual gas pressure could be reduced even further.

We have discussed the importance of the vacuum quality and anticipated upgrades around lines 309ff. We have now added the following, additional sentence at the end of this paragraph to reflect your comment: ***“These improvements will be particularly beneficial for LPT studies of elements with lower electron affinity, where the anions may be more prone to neutralization upon collisions with residual gas particles.”***

Some editorial considerations for the authors:

Abstract: the explicit mention of the five-orders-of-magnitude reduction in sample size is mentioned in the same sentence as the improvement of sensitivity, qualified as “greatly improved”. Why not also quote the (greater than) three-order-of-magnitude gain in sensitivity, nicely justified in Fig. 4? I might also suggest that the authors distinguish the additional gain brought by the ion “recycling” of the lower-power lasers, rather than grouping in the same sentence (which slightly diminishes the impact). For example: “Moreover, the additional exposure time allows the use of lower-power CW lasers that reduce uncertainties due to broadening.”

We modified the abstract as suggested, see below. The changes are indicated in bold.

*“The electron affinity (EA), the energy released when a neutral atom binds an additional electron, is a fundamental property of atoms which is governed by electron-electron correlations and is strongly related to an element's chemical reactivity. However, conventional techniques for EA determination lack the experimental sensitivity to probe very scarce samples. As a result, the EA for the heaviest elements of the periodic table is entirely uncharted. Here, we present a novel technique to determine EAs through Laser Photodetachment Threshold Spectroscopy, performed in an electrostatic ion beam trap to increase the samples' exposure to laser photons and, **thus, improving the experimental signal sensitivity by more than three orders of magnitude. Moreover, the additional exposure time allows the use of lower-***

power continuous-wave narrow-band lasers that reduce uncertainties associated with broadening effects induced by the laser bandwidth. By applying this technique, we measure the EA of ^{35}Cl to be $3.612720(44)\text{ eV}$, achieving state-of-the-art precision while employing five orders of magnitude fewer anions. The demonstrated sensitivity paves the way for systematic EA measurements across isotopic chains - including isotope shifts and hyperfine splittings - and ultimately for the first direct determination of electron affinities in superheavy elements.”

Introduction – short suggestion for rewording the first sentence: “Unveiling the intricate electronic structure of atoms, including understanding how their attributes define the macroscopic chemical properties, was one of the greatest scientific achievements of the last century.”

I also suggest modifying: “experimentally entirely unknown” to read: “experimentally unknown”.

We changed as suggested to “Unveiling the intricate electronic structure of atoms, **including understanding how their attributes define the macroscopic chemical properties**, was one of the greatest scientific achievements of the last century.”

“Despite its fundamental importance, the EAs of several elements -- especially those that are rare, heavy and radioactive -- **are experimentally unknown**”^{~\cite{Ning2022}}.”

Reviewer #2 (Remarks to the Author):

The authors present the application of a new technique to improve the sensitivity of laser photodetachment threshold (LPT) spectroscopy. Currently LPT is one of the most precise method in determining the electron affinity of negative ions. However, due to the low cross section of the non-resonant photodetachment process the mainly used single pass LPT is hindered when determining the EA of rare or exotic species. The logical consequence is to go for a multi-pass measurement where the precious ions are probed multiple times and therefore increasing the sensitivity. Inspired from storage ring experiments the authors use a Multi-Reflection Rime Of Flight apparatus for this purpose. Keeping in mind that the previous storage ring measurements were performed with much higher ion beam intensities I disagree with the statement in the introduction arguing that storage ring measurements are not therefore not compatible with RIB facilities. Facilities like the ESR at GSI show that even single ion sensitivity can be achieved using storage rings, but undeniably under large infrastructure effort.

We fully agree with the reviewer on the remarkable achievements of storage-ring measurements, particularly those involving radioactive ions at the ESR at GSI. The ability to reach single-ion sensitivity in mass measurements, even for very short-lived radioactive species, is truly impressive. However, according to our knowledge, this single ion sensitivity has exclusively been achieved for mass measurements at ESR, not for any laser spectroscopic investigation. For example, in the recent fluorescence-based laser spectroscopic publication at the ESR (M. Horst et al., Storage-ring laser spectroscopy of accelerator-produced hydrogen-like $^{208}\text{Bi}^{82+}$, Nat. Phys. 21, 1057-1063, 2025) the authors write that their measurement was carried out utilizing 100-1000x fewer ions than the previous measurement that had been performed with 10^8 ions circulating in the ESR. Despite this remarkable advance, this is still far away from single ion sensitivity. Moreover, injecting negatively-singly-charged ions into the ESR would require major technical upgrades, if possible at all. Hence, no photodetachment

measurement has to our knowledge been performed thus far in the ESR. Coupling a low-energy storage ring like DESIREE to a radioactive ion beam facility and performing photodetachment measurements would be feasible in principle though. However, all the photodetachment measurements at the DESIREE storage ring are performed with ion intensities several orders of magnitude higher than what is available for exotic species thus far (see lines 91ff). While we believe that single-ion sensitivity for EA measurements could, in principle, be achieved in a storage ring, it has not been demonstrated thus far. Hence, no changes to the manuscript text were made.

To present the validity of their approach the authors present a measurement of the EA of Cl-35 using the MR-TOF technique. The results gained are thoroughly analysed and the conclusion drawn are sound. The proposed method can have a significant impact on future electron affinity determination of rare and exotic species. Being a key experimental accessible parameter of negative ions the electron affinity gives important insight in further properties of rare species like e.g. their chemical behaviour. The data and methodology presented in the supplementary material describes reasonably well the experimental methods and the modelling of the parameters specially focused on the used MR-TOF device. A comprehensive discussion of the achieved sensitivity is given. The list of references is exhaustive and all the relevant previous works relevant for the current article have been cited.

Thank you for the positive evaluation of our work!

A suggested improvement would in my opinion be a more comprehensive discussion of the pros and cons of the presented technique in comparison to using storage rings. Both methods gain from the use of multi-pass instead of the low sensitivity of a single pass experiment. However, both complicating effects of 'continuous Doppler shift' and 'blueshift depletion' are intrinsic to a MR-TOF device and non-existent in a storage ring. The manuscript would gain from such a discussion and emphasise why the authors claim that their method is superior.

As we have discussed above, no photodetachment measurement involving only a few ions has, to our knowledge, been realized in a storage ring to date. While we acknowledge that storage rings are free from continuous Doppler shift and blueshift depletion—both of which are inherent to MR-ToF devices—we also note that the practical implementation of photodetachment measurements with only small ion intensities in storage rings remains unproven. We appreciate the reviewer's suggestion to more thoroughly discuss the comparative merits of our method relative to storage ring approaches. However, due to the strict word limit imposed by Nature Communications, we have focused this manuscript on the demonstrated capabilities and performance of our method, rather than on a detailed discussion of hypothetical, yet-to-be-demonstrated alternatives.

Reviewer #3 (Remarks to the Author):

This manuscript, "Enhanced Sensitivity for Electron Affinity Measurements of Rare Elements," introduces a significant advancement in measuring Electron Affinity (EA), addressing the critical challenge of measuring EA of rare and heavy elements, a fundamental atomic property that remains uncharted due to insufficient experimental sensitivity of conventional techniques. The authors combined Laser Photodetachment Threshold (LPT) spectroscopy with a Multi-Reflection Time-of-Flight (MR-ToF) device

to create a highly sensitive new method. The technique involves generating and injecting chlorine anions ($^{35}\text{Cl}^-$) into the MR-ToF device, where they are trapped for about 0.5 seconds (60,000 revolutions). A continuous-wave laser repeatedly probes the anions during this time. The paper demonstrates a three-order-of-magnitude improvement in signal sensitivity over conventional methods, using approximately five orders of magnitude fewer anions and less laser power. This breakthrough is poised to have a significant impact on atomic physics, nuclear physics, and the chemistry of super-heavy elements. For unstable isotopes, such as the heaviest elements, conventional measurement methods are not applicable. That's why the technique reported in this manuscript will pave the way for atomic sciences in the study of superheavy elements.

I recommend minor revisions for publication, which will help the reader's understanding. The minor revision list is as follows.

Thank you very much for this highly positive review! We have addressed all the minor revisions, see below.

Minor Comments

- Line 120: The statement "A continuous beam of chlorine anions containing both stable isotopes of chlorine, ^{35}Cl and ^{37}Cl , is produced..." is followed by "a deflector which isolates the ^{35}Cl isotope by time-of-flight selection". Please clarify if ^{37}Cl is ever actually used, or if its initial mention is purely contextual.

Its initial mention is purely contextual. Since there is a small isotope shift between ^{35}Cl and ^{37}Cl we made sure that we only have ^{35}Cl stored in the MR-ToF device, to measure exclusively the electron affinity of ^{35}Cl . We added the following text in bold to the manuscript: "*In this process, the anion bunches pass through two high-voltage switching elements: a pulsed drift tube, which adjusts the anions' kinetic energy, and a deflector which isolates the ^{35}Cl isotope by time-of-flight selection **ensuring that the measurement pertains solely to the electron affinity of ^{35}Cl .***"

- Line 132: Regarding "collision-induced detachment with residual gas particles at a pressure of 3×10^{-8} mbar," it would be beneficial to provide context on whether this pressure is typical for ion beam traps or a specific challenge for the MIRACLS apparatus, e.g., compared to storage rings.

As already noted in our response to Reviewer 1, the vacuum quality of the specific ion-trap system used in this work can be improved and is by no means an inherent limitation of the MIRACLS approach. The present MR-ToF device is a first-generation instrument and we are aware of several limitations in the construction of this specific MR-ToF apparatus (e.g. components with relatively large outgassing rates are used to mount the MR-ToF mirror electrodes). We hence foresee that the vacuum in a future setup built dedicated for only photodetachment measurements can be improved by two orders of magnitude. In case of a cryogenic MR-ToF device, the residual gas pressure is expected to be even further reduced.

We made the following change in bold to lines 132ff: "*As stored anions are neutralized, either by laser photodetachment or collision-induced detachment with residual gas particles at a pressure of 3×10^{-8} mbar **in the present MR-ToF device**, they escape as neutral atoms and are detected by a MagneTOF Mini Detector from ETP Ion Detect placed outside the ion trap.*"

- Line 143: "Photoelectrons that are emitted when laser (stray) light hits the detector" are mentioned as a background source. Given that the detector is "slightly displaced off-axis" (Line 138) "to avoid direct laser incidence," please clarify the nature and significance of this "stray light" background relative to the detector's dark counts.

We added the text in bold: *"This component of the background, originating from detector dark counts and photoelectrons that are emitted when laser (stray) light hits the detector, accounts for about 3% of the total counts per cycle. **The detector dark counts amount to ≈ 0.4 counts per cycle and the laser-induced background counts are ≈ 0.2 counts per cycle for each milliwatt of laser power.**"*

- Line 166: The statement " $^{35}\text{Cl}^-$ anion has a pure $3p^6 1S_0$ configuration which does not exhibit any fine nor hyperfine splitting" refers to the anion. However, Line 170 states, "the ground state of the remaining neutral atom has a configuration of $3p^5 2P_{3/2}$, which splits into four hyperfine m states...". To avoid confusion, explicitly specify that the lack of fine/hyperfine structure refers to the anion ground state before photodetachment.

We added the text in bold: *"Finally, **after the photodetachment process** the ground state of the remaining neutral atom has a configuration of ..."*

- Line 217: The description " $\Phi = P/E$ is obtained from the photon energy E and laser power P in each measurement, typically around 2 mW". For clarity, it would be more precise to write $\Phi = P / (h\nu)$ or P/E_{photon} to indicate that E refers to photon energy explicitly.

We agree that this would be better, however the photon energy E is already introduced in equation 2 and reused at various positions in the methods section. We would hence need to exchange the variable name at all positions, which we would like to avoid (also to be consistent with other photodetachment literature).